# Colour Stabilisation of Surface of Four Thermally Modified Woods with Saturated Water Vapour by Finishes

**DOI:** 10.3390/polym13193373

**Published:** 2021-09-30

**Authors:** Zuzana Vidholdová, Gabriela Slabejová

**Affiliations:** 1Department of Wood Technology, Faculty of Wood Sciences and Technology, Technical University in Zvolen, T.G. Masaryka 24, 96001 Zvolen, Slovakia; 2Department of Furniture and Wood Products, Faculty of Wood Sciences and Technology, Technical University in Zvolen, T.G. Masaryka 24, 96001 Zvolen, Slovakia; slabejova@tuzvo.sk

**Keywords:** alder, beech, birch, colour, maple, surface finish, thermally modified wood

## Abstract

This paper deals with the influence of the type of transparent surface finish on the change of colour of the surfaces of native wood, and thermally treated wood, with saturated water vapour. In the experiment, alder, European beech, Paper birch, and Norway maple wood were thermally treated at a temperature of 135 °C under saturated water vapour for six hours. Three various types of surface finishes (acrylic-polyurethane, polyacrylic and aldehyde resin, and alkyd resin) were applied onto the wood surfaces. The colours of the surfaces in the system, CIE *L*a*b** (lightness, coordinates *a** and *b**, chroma and hue angle), were measured during finishing and natural ageing behind glass windows in an interior, over a period of 60 days. The results show that the changes in the yellowness index, and the total colour differences after the application of individual surface finishes to wood species, changed because of sunlight exposure. Moreover, it is clear that different wood finishes behaved differently on all of the wood species. An analysis is presented in this paper.

## 1. Introduction

From an environmental perspective, thermally based modification processes are extremely interesting because they are implemented to improve the properties of wood, to produce new materials, and to provide the form and functionality desired by engineers, architects, and designers without changing the eco-friendly characteristics of the materials [1]. The thermo-hydro treatment of wood is one category of thermally based modification in which water, in the form of liquid water, saturated water steam, or a saturated humid air environment, acts as a modification agent in this process.

The thermal treatment of wood with saturated water vapour is traditionally used in the woodworking industry, for example, in the manufacture of furniture components with solid wood bending, lamination of veneer (laminated bending), interior tiling, and flooring. The thermally-based modification treatment is also accompanied by the chemical reactions of the cell-wall components (polysaccharides, lignin, and extractives), which cause changes in the colour of the wood [1,2,3,4,5,6,7,8].

Sunlight is the major cause of damage to a number of materials, including wood, coatings, and other organic materials. The type of damage, such as a loss of gloss, chalking, elasticity, adhesion, and colour change, varies depending on the material sensitivity and the spectrum of sunlight. The mechanisms of wood photodegradation have been investigated, and it appears that lignin is the key structure because this component is able to absorb the ultraviolet light (UV) and the visible light region because of its chromophoric groups. The main chromophoric groups in native wood lignin are various kinds of ring-conjugated ethylene groups, such as cinnamyl alcohol, and the cinnamaldehyde groups together with the carbonyl groups, such as α—carbonyl groups and quinones [9]. The absorption of light by these groups initiates the formation of free radical species and these free radicals react with oxygen to form the chromophore groups, such as the carbonyl and carboxyl groups, that ultimately lead to the discolouration of the wood [10,11,12,13,14,15,16,17].

Coatings can give wood materials the desired aesthetic properties, such as colour and gloss, but are also generally essential in protecting wood from environmental influences, such as moisture, radiation, biological damage, or damages of mechanical or chemical origins. This applies to both interior uses (like furniture) [18,19,20,21], and exterior applications [22,23,24,25]. From the point of view of customers, the aesthetic appearance of wood coating is the main purchasing factor.

Transparent or clear finishing is designed to enhance the stability of the wood surface and maintain the natural aspects of the wood, such as the colour, grain, and texture, for a long time. Transparent finishing films perform badly on wood surfaces during interior or exterior exposure. In fact, these types of coatings cannot absorb UV light, and attend the wood surface [26]. This phenomenon leads to the photodegradation of the wood substrate. A visible colour change in wood is the first sign of its chemical modification when exposed to light, even in diffuse indoor light conditions. Changes in the colour of the wood surface after applying a transparent coating material are the result of an interaction of the colour of the coating film with the colour of the wood surface. Various transparent finishes produce different colours in wood surfaces [27]. The change in colour due to the surface finish is an interaction of the changed wood colour and the colour of the coating film itself. It is generally known that coating films exposed in interiors turn yellow under the influence of sunlight. The most commonly adopted UV protection technology is to use UV protective substances that are admixed into the coating material. However, this degradation has not been inhibited absolutely [16,19,24,25].

The objective of this paper is to characterise the aesthetic performance of four wood species thermally treated with saturated water vapour under specific interior conditions, and to assess the possibility for its improvement with three transparent surface finishes.

## 2. Materials and Methods

### 2.1. Sample Preparation and Finishing

In this study, alder (*Alnus glutinosa* (L.) Gaertn), European beech (*Fagus sylvatica* L.), Paper birch (*Betula papyrifera* Marsh), and Norway maple wood (*Acer pseudoplatanus* L.) were used. Samples of 15 × 100 × 250 mm^3^ (radial × tangential × longitudinal) were prepared from both TM boards and unmodified (native) boards with saturated water vapour at a 135 ± 2.5 °C temperature for 6 h. The parameters for the modification process for the selected wood species are described in more detail in the works [4,5,6,7]. The samples were prepared from six-month air-conditioned and sanded boards (last sand grit P 180, sanded first in the transversely and following in the longitudinal direction). The samples had 3 to 8 growth rings per cm, they were free from defects, and the growth ring orientation to the tested surface was 5° to 45°.

In the study, three transparent surface finishes (Table 1) for interiors were applied to the samples based on the producer’s recommendations:•One-component water-based acrylic-polyurethane dispersion surface finish, Aqua TL-412-Treppenlack/50. It is recommended for use on solid wood, veneers, wooden stairs, and living room and bedroom furniture. It is an abrasion-resistant finish with a high solids content.•Two-component surface finish with polyacrylic and aldehyde resin, PUR SL-212-Schichtlack/30. It is recommended for use on solid wood, veneers, tables and worktops, and kitchen and bathroom furniture. It is highly scratch-resistant and full-built.•Single-component wood sealer with alkyd resin, HWS-112-Hartwachs-Siegel/clear. It is recommended for use on furniture, tables and worktops, bathroom and sauna elements, floors and stairs, cork floors, and bamboo components.

After application, samples were stored for 14 days at 23 °C and 50% relative humidity (RH) in a dark room to ensure film formation, sufficient hardening, and solvent evaporation.

### 2.2. Natural Sunlight Ageing

The natural sunlight irradiation was carried out between July 2020 and September 2020 for a total of 60 days when the interior temperature varied from 20 to 25 °C, and RH varied from 50% to 55%. The daily average of total solar power density was between 336 and 535 W·m^−2^ in Zvolen, Slovakia. The coated and uncoated specimens were stored in a room inside, behind a glass window (thermal-insolation double glazing with U-factor 1.1 W·m^−2^·K^−1^ with west direction). The geographical data for Zvolen are: longitude 19°07′03″ East; latitude 48°34′15″ North; and an altitude of 283 m.

### 2.3. Colour Analyses

The colour parameters of the tested samples were measured before and after the coating or ageing using a Color Reader CR-10 (Konica Minolta, Osaka, Japan). The device was set to an observation angle of 10°, with d/8 geometry, and a D65 light source. The colour changes of the sample surfaces were monitored for 60 days at 10 given positions for each sample.

The CIELAB parameters, lightness *L**, coordinates *a** and *b**, chroma *C*_ab_*, and hue angle *h_ab_* were measured. The differences between the two stimuli were calculated as follows (according to the standard ISO 7727-3 [29]):(1)$$\Delta L^{*}=L_{1}^{*}-L_{0}^{*},$$
(2)$$\Delta a^{*}=a_{1}^{*}-a_{0}^{*},$$
(3)$$\Delta b^{*}=b_{1}^{*}-b_{0}^{*},$$
(4)$$\Delta C_{ab}^{*}=C_{ab,1}^{*}-C_{ab,0}^{*},$$
(5)$$\Delta h_{ab}=\Delta h_{ab,1}-\Delta h_{ab,0},$$
where the index “0” means the average values that represent the colour coordinate “before” (surface finishing or ageing test = exposure time 0 days), and index “1” means the average values that represent the colour coordinate “after” (surface finishing or ageing test = exposure time 10, 20, 30, and 60 days).

The yellowness index is a number calculated from spectrophotometric data that describes the change in the colour of the test samples. The yellowness index *YI*, and a change in the yellowness index ∆*YI*, were subsequently calculated as (according to standard ASTM E313-96 [30]):(6)$$YI=\frac{1.3013\cdot X_{CIE}-1.1497\cdot Z_{CIE}}{Y_{CIE}}\cdot 100,$$
(7)$$\Delta YI=YI_{1}-YI_{0},$$
where *X_CIE_*, *Y_CIE_*, and *Z_CIE_* are the tristimulus values. The *YI_0_* means the average values of the yellowness index before the ageing test (exposure time 0 days), and the *YI_1_* means the average values of the yellowness index after the ageing test (exposure time 10, 20, 30, and 60 days). By this calculation, a positive (+) ∆*YI* indicates increased yellowness, and a negative (−) ∆*YI* indicates decreased yellowness, or increased blueness.

The total colour difference, ∆*E**_ab_, was subsequently calculated as the Euclidean distance between the points representing them in the space (according to the standard ASTM D2244-16 [31] and ISO 7727-3 [29]):(8)$$\Delta E_{ab}^{*}=\sqrt{\Delta L^{*2}+\Delta a^{*2}+\Delta b^{*2}},$$

To demonstrate the colour change of the coated wood surfaces, the Scanner HP LaserJet 1536dnf MFP was used before and during the exposure of the specimens to natural sunlight ageing.

### 2.4. Statistical Evaluation

The MS excel 2013 and statistical software STATISTICA 12 were used to analyse and present the collected data on colour parameters. Descriptive statistics deal with basic statistical characteristics—arithmetic mean and standard deviation, and an analysis of variance (ANOVA) at an = 0.05 significance level.

## 3. Results and Discussion

In this complex review, the effects of the wood species, wood treatment, finishing, and exposure time, and their interactions on the measured colour parameters after natural sunlight ageing, were statistically evaluated (see Table 2).

A statistically significant effect (with a *p*-value less than 0.05) was found for the effects of wood species, wood treatments, and finishes for all the measured colour parameters. However, the exposure time had a statistically insignificant effect on the redness parameter. It should be noted that there is also a significant interaction effect between the finishing and the wood treatment (F × WT) for all the measured colour parameters.

The results of this study will be presented as per the above-mentioned view.

### 3.1. Initial Colour of Wood and Effect of Finish Application

The colour parameters (*L**, *a**, *b**, *C*_ab_*, and *h_ab_*) of native (original), and thermally modified wood, with saturated water vapour before coatings, are given in Table 3. 

All of the TM wood species differed mainly in lightness *L** compared to the native woods; it were higher, from 15.85 on Norway maple, to 19.95 on alder. Moreover, all woods had the colour parameters *a** and *b** in a positive sphere of distribution. The highest value of chroma *C*_ab_* was observed for Norway maple wood and, at the same time, the highest difference was found for this wood. The hue angle *h_ab_* was in the first quadrate (ranges for wood between 0°and 90°). All woods had a yellow hue that was predominant over the red hue, most typically for native Paper birch. On the contrary, for the TM alder, the red shade was slightly dominant. The changes in the colour of all wood species resulting from thermal modification by saturated water vapor are considered to be of a permanent nature and irreversible. The irreversibility of the changes in the colour of woods is confirmed by the differences in the lignin-carbohydrate complex of the TM wood, as well as native wood, and by the presence of monosaccharides, organic acids, and the basic structural elements of the guaiacyl-syringyl lignin in the condensate [1,4,5,6,7,8]. Irreversible changes in the colour of the TM wood expand the opportunities for the use of these four woods in the fields of construction-joinery, and construction art and design.

The wood species were then coated with selected transparent finishes. Figure 1 shows the change of their colour parameters (∆*L**, ∆*a**, ∆*b**, ∆*C*_ab_*, and ∆*h_ab_*).

The differences in the lightness ∆*L** was always negative; the surfaces darkened on both the TM and the native alder, as well as on the European beech, Paper birch, and Norway maple wood. The wood surfaces with the alkyd surface finish darkened the most, except on the TM and native alder wood, and the TM Norway maple wood.

Positive changes ∆*a** were on all surface finishes on both wood surfaces, except on TM Paper birch and native Norway maple wood, after applying the one-component surface finish with acrylic/polyurethane dispersion. The *a** coordinate was in the red area and, after applying the surface finishes, it was even more pronounced towards the red.

Positive changes in ∆*b** were apparent on both wood types (TM and native wood). The *b** coordinate after the application of the surface finishes was more pronounced towards the yellow. The greatest ∆*b** occurred after applying the alkyd surface finish.

The difference in chroma ∆*C*_ab_* was positive on both wood types. The tested surfaces showed increasing chroma, mostly after applying the surface finish with alkyd resin. Prior to the application of the coating materials, the surfaces tended to be more yellow than red. The differences in the hue angle ∆*h_ab_* after application of the coating materials shows that the colour of the surfaces was even more pronounced towards yellow than towards red, except on TM Norway maple and alder wood, with a two-component surface finish with polyacrylic resin and aldehyde resin. Similar results are presented in [27]. The authors researched the changes in the colour of the surfaces of Norway maple wood and TM wood after the application of surface finishes. The transparent coatings applied on Norway maple, on both TM woods with saturated water vapour, and on native wood, caused a negative difference in lightness and positive differences in the chroma ∆*C*_ab_* and the hue angle ∆*h_ab_*.

### 3.2. Effects of Natural Sunlight Ageing on Differences in ∆L*, ∆a*, ∆b*, ∆C*_ab_, and ∆h_ab_

The trend of discolouration was observed on all of the coated and uncoated samples during the 60 days of indoor sunlight exposure under glass windows. The differences in ∆*L**, ∆*a**, ∆*b**, ∆*C*_ab_*, and ∆*h_ab_* are presented graphically in Figure 2, Figure 3, Figure 4 and Figure 5.

The effect of natural sunlight ageing on native wood species showed a difference among the four wood species compared to the effect on TM woods. In Figure 2a, Figure 3a, Figure 4a and Figure 5a, it can be seen that the differences in +∆*a** (redness) increased slowly in comparison to the differences in +∆*b** (yellowness) on the native wood species, and both changes ∆*a** and ∆*b** increased with irradiation time. The rate of change was higher in the first stages of irradiation. In the first ten days, the exposure to sunlight of the two wood species showed marked yellowing and some redness. This situation was noted for Norway maple, where the value of ∆*b** was eight times higher than ∆*a**, and for alder it was 1.4 times higher. Paper birch showed a difference, where −∆*b** was five times higher than ∆*a**. On the other hand, European beech showed a yellower and even more intense red, and the value of ∆*a** was 1.4, and the value of +∆*b** was 0.6. After 60 days of exposure, the ratio of yellowness and redness ∆*b**/∆*a** decreases in order of Norway maple (3.9) > Paper birch (3.4) > European beech (2.6) > alder (1.8). In the case of the thermally treated wood (see Figure 2d, Figure 3d, Figure 4d and Figure 5d), it can be seen that the difference −∆*a** (less red) has changed differently compared to the difference ∆*b**. In the first ten days of sunlight exposure, only alder showed yellowing +∆*b** and less reddish −∆*b**, with the ∆*b**/∆*a** at 2.3. The other three wood species showed tendencies towards being less red −∆*a** and less yellow −∆*b**, and ∆*b**/∆*a** decreases in order of Paper birch (1.1) > European beech (0.8) > Norway maple (0.5). After 60 days of exposure, all of the thermally treated wood species showed a similar tendency to be yellower +∆*b** and to be less red −∆*a**, with the ratio of ∆*b**/∆*a** decreasing in order of alder (2.8) > Paper birch (2.5) > European beech (2.4) > Norway maple (2.3). Similar behaviour was observed in a study by Pandey [32] for photoinduced changes in uncoated softwood and hardwood. Salcă and Cismaru [33] reported that the *a** and *b** colour coordinates of black alder veneers had an increasing tendency with increasing exposure time, which signifies a colour darkening under sunlight radiation that penetrates the window glass. The colour coordinates were more pronounced during the first month of exposure, and slower within the next one. Tolvaj and Mitsui [34] showed the rapid colour change of beech, black lotus, Japanese cedar, and spruce during the initial period of irradiation, and the rate of colour change decreased upon prolonged irradiation. They also report that the wood surfaces, which were irradiated with simulated indoor sunlight exposure using a mercury lamp, discoloured more rapidly than those exposed to natural sunlight. Miclečić et al. [35] reported that, in the first ten days of sunlight exposure, the surfaces of uncoated thermally modified ash, beech, and hornbeam samples discoloured slowly compared to uncoated native samples.

The differences in chroma ∆*C*_ab_* in correlation with the difference in lightness ∆*L** on native wood are given in Figure 2b, Figure 3b, Figure 4b and Figure 5b. For the first ten days, the native alder, European beech, and Norway maple surfaces showed a positive difference in chroma. Then, for all the wood species, the difference started rising positively in value. At the same time, the difference in the lightness was negative, meaning darker. However, the cases of the TM European beech, Paper birch, and Norway maple wood (Figure 2e, Figure 3e, Figure 4e and Figure 5e) showed a negative difference in chroma and a positive difference in lightness, meaning surfaces become lighter. A decrease in the lightness of native and thermally modified wood species is in compliance with the knowledge about the changing colour of wood, and its darkening, during the processes of natural sunlight weathering mentioned in [15,32,33,34,35].

The differences in the hue angle value were lower for the native wood species than for the TM wood samples, as seen in Figure 2c, Figure 3c, Figure 4c and Figure 5c, and Figure 2f, Figure 3f, Figure 4f and Figure 5f. The tendency difference of the hue angle increase was similar to the tendency of the yellowness and redness increase, and the differences within the first ten days of the test for the individual types of wood species. For interpretation, −∆*h_ab_* represents a tendency to be red, and +∆*h_ab_* represents a tendency to be yellow.

Among all the coating systems, lower colour differences ∆*a**, but higher ∆*b** among exposure to natural sunlight, were measured in native samples coated with one-component water-based acrylic-polyurethane dispersion (Figure 2a, Figure 3a, Figure 4a and Figure 5a). Beech wood achieved this trend after 60 days of sunlight exposure. Moreover, the finish with one-component alkyd resin showed the lowest ∆*b** difference but, at the same time, a high colour difference of ∆*a**, especially in the case of alder wood. After 60 days of exposure, the reverse ratio ∆*a**/∆*b** decreased in order of: alder (3.4) > Norway maple (3.0) > Paper birch (2.7) > European beech (1.1). The largest changes in the parameters ∆*a** and ∆b*** were found for the finishes of two component polyacrylic and aldehyde resins on all four wood species. When the thermally modified wood species were exposed to sunlight, the changes in the ∆*a** and ∆*b** parameters of all three finishes were similar to those observed in the native wood species (Figure 2d, Figure 3d, Figure 4d and Figure 5d).

For example, also in the study by Durmaz et al. [36], it was found that waterborne acrylic coatings can improve the weatherability of wood−plastic composites and minimize colour changes—the lightness difference ∆*L**. In the studies, Gurleyen et al. [37], and Ayata et al. [38], polyacrylic-based resin-coated wood showed a slightly lower decrease in lightness. The decrease in lightness was probably due to the formation of coloured degradation products during the heat treatment. There was a slight increase, followed by a decrease in redness (lower *a**), and a clear decrease in the yellow tone (*b**).

### 3.3. Effects of Natural Sunlight Ageing on Effects on Difference in Yellowness Index

As shown above, intensive yellowing with an increasing ageing time was valuated as the difference of ∆*b**. When the native and thermally modified wood species are exposed to sunlight, the changes in the difference in the yellowness index parameter ∆*YI* of all three finishes were different (Figure 6). The highest ∆*YI* parameter was noted for two-component polyacrylic and aldehyde resin on all four types of wood. With increasing ageing time, all values of the native wood species were positive and indicated increased yellowness. The rates of change were higher during the first 30 days of ageing. However, in the case of two-component polyacrylic and aldehyde resins on thermally treated wood species, the negative (−) ∆*YI* indicates reduced yellowness, or increased blueness, during the first 30 days of ageing, and then between 30 and 60 days of aging, the coating tended to turn yellow. The other two one-component finishes showed a similar tendency towards changing the difference in the yellowness index ∆*YI* parameter, on native wood species with positive values, indicating increased yellowness, but on thermally modified wood species with negative values, indicating decreased yellowness. The finish with a water-based acrylic-polyurethane dispersion on native alder wood showed the difference when +∆*YI* (increased yellowness) was 5.7 times lower than ∆*YI* in the case of a two-component polyacrylic and aldehyde resin.

When evaluating TM and native alder wood after 60 days of exposure to daylight, the lowest yellowing was shown by the one-component water-based acrylic-polyurethane dispersion surface finish. For the TM European beech wood, the lowest yellowing was shown by the single-component wood sealer with an alkyd resin surface finish, and the one-component water-based acrylic-polyurethane dispersion surface finish. Although for the native one, the lowest yellowing was shown by the single-component wood sealer with an alkyd resin surface finish. In case of the TM Paper birch wood, the lowest yellowing was shown by the single-component wood sealer with an alkyd resin surface finish, and the one-component water-based acrylic-polyurethane dispersion surface finish as well. For the native one, the lowest yellowing was shown by the one-component water-based acrylic-polyurethane dispersion surface finish. And finally, for the TM Norway maple wood, the lowest yellowing was shown by the one-component water-based acrylic-polyurethane dispersion surface finish. For the native one, a similar reduction in yellowing was shown by the single-component wood sealer with an alkyd surface finish, and by the one-component water-based acrylic-polyurethane dispersion surface finish. The discoloration of all the transparent coated systems resulted mainly from the photoyellowing of the underlying native or thermally modified wood. The findings in this research agree with our publication [21], and with other authors [39,40,41].

### 3.4. Effects of Natural Sunlight Ageing on Total Colour Difference ∆E*_ab_

The total changes in colour due to exposure to sunlight for the wood species studied, expressed by the calculated colour difference values Δ*E*_ab_*, are presented in Figure 7.

The highest Δ*E*_ab_* value after 60 days of exposure was registered for two-component polyacrylic and aldehyde resins on the native Norway maple (6.3 and 14.00, respectively), and the lowest value was for the one-component water-based acrylic-polyurethane dispersion on alder native wood (2.4). For the thermally modified wood species, the Δ*E*_ab_* values varied between 5.1 (finish with alkyd resin on Paper birch), and 10.6 (two-component polyacrylic and aldehyde resin on Norway maple). All experimental colour data (Figure 7) indicate that photodegradation was initiated very quickly, with detectable effects after 10 days, and this evolved as the exposure time increased (up to a total of 60 days). In addition, it is clear that different wood species behave differently. The total colour difference Δ*E*_ab_* shows a systematic trend towards higher values with increasing irradiation time. However, their further development at longer irradiation times cannot be predicted from this measured data, and more research at longer exposure times is needed.

The findings in this research agree with the publications of other authors for native wood surfaces (e.g., [15,32,33,34,42]), the surfaces of thermally modified wood [21,35], and clear coated wood, although most of the reported research [21,35,36,37,38,39,40,41,43,44] used simulated indoor sunlight exposure or other accelerated weathering methods. In these studies, the dependence curves showing the colour change had an initial period of severe increase that was maintained at a practically constant value with further exposure. However, important changes occurred after short exposure times.

## 4. Conclusions

The colour and visual dimension of wood is a very important feature, and its maintenance is required during the long-term life of wood products. Sunlight is the primary cause of damage to wood and its coatings in interiors. This study has shown that TM wood species—alder (*Alnus glutinosa* (L.) Gaertn), European beech (*Fagus sylvatica* L.), Paper birch (*Betula papyrifera* Marsh), and Norway maple wood (*Acer pseudoplatanus* L.)—differ considerably from unmodified wood with regard to the colour change due to natural sunlight exposure behind the glass window. The native wood species showed intensive yellowing and a higher value of the total colour difference than thermally modified wood species.

Transparent or clear finishing is designed to enhance the stability of the wood surface and maintain the natural aspect of the wood. This study has shown that coated wooden surfaces were susceptible to discolouration, even though they had been coated with the transparent finishes (acrylic-polyurethane, polyacrylic and aldehyde resin, and alkyd resin). The discolouration of the finishing systems resulted mainly from the photoyellowing of the underlying native or thermally modified wood and increased with the irradiation time.

From the point of view of preserving the original colour with reduced yellowing due to the action of sunlight on the native and TM European beech wood, it has been shown that the use of a one-component wood sealant with an alkyd resin coating is the best choice. In the case of lighter types of natural wood, such as alder, Paper birch, and Norway maple, the reduced yellowing and discoloration were characterized by a surface treatment with a one-component water-based acrylate-polyurethane dispersion. Although this finish was applied on thermally modified alder, Paper birch, and Norway maple, the yellowing was reduced. In terms of reducing the overall colour difference, the lowest change was recorded for the one-component alkyd-coated wood sealant on thermally modified Norway maple wood. However, the one-component water-based acrylic-polyurethane dispersion surface finish caused intensive yellowing and, in many cases, caused the highest colour difference on all native and thermally modified wood species and is, therefore, unsuitable from this point of view.

## Figures and Tables

**Figure 1 polymers-13-03373-f001:**
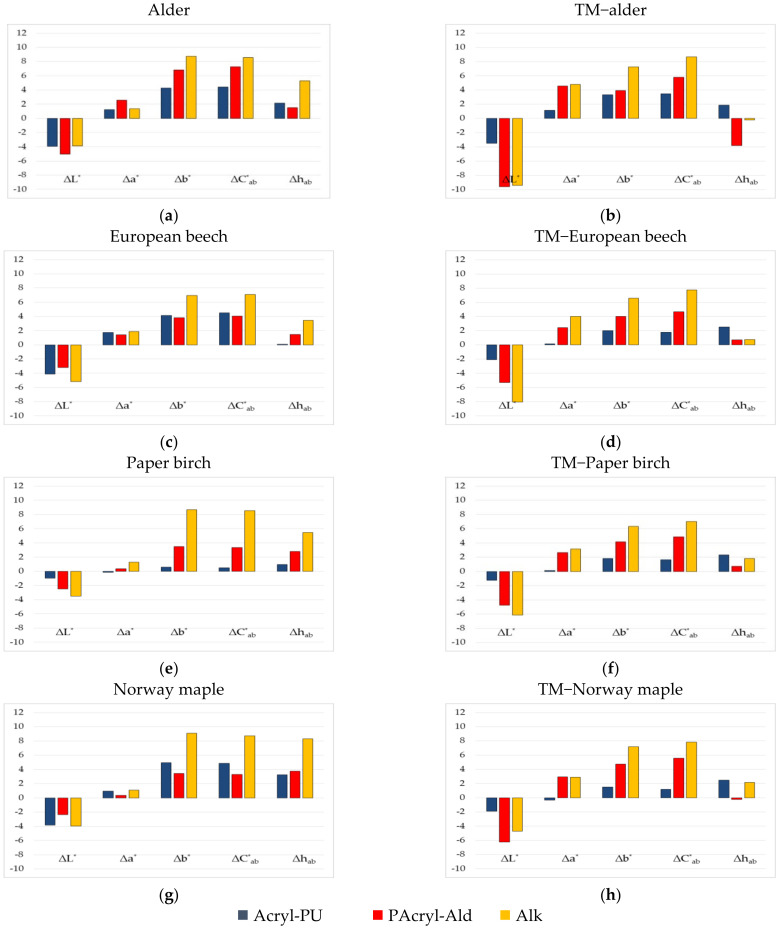
Differences in coordinates ∆*L**, ∆*a**, ∆*b**, ∆*C*_ab_*, and ∆*h_ab_* after surface finishing: native (**a**,**c**,**e**,**g**), and TM with saturated water vapour (**b**,**d**,**f**,**h**), alder, European beech, Paper birch, and Norway maple wood.

**Figure 2 polymers-13-03373-f002:**
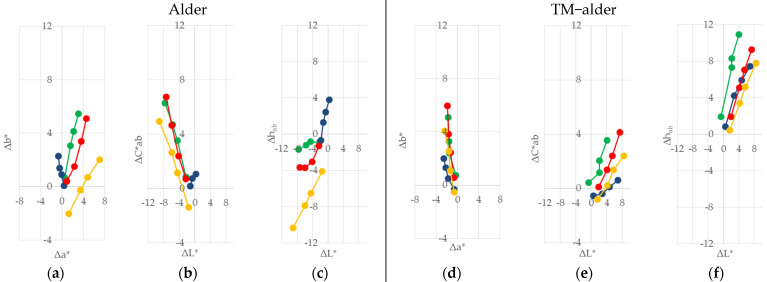
Changes in ∆*L**, ∆*a**, ∆*b**, ∆*C*_ab_*, and ∆*h_ab_* after the exposure to indoor sunlight of coated and uncoated native (**a**–**c**), and thermally treated (**d**–**f**), alder wood for 60 days. Note for Figure 2, Figure 3, Figure 4 and Figure 5: green = uncoated wood; blue = coated with Acryl-PU; red = coated with PAcryl-Ald; and yellow = coated with Alk.

**Figure 3 polymers-13-03373-f003:**
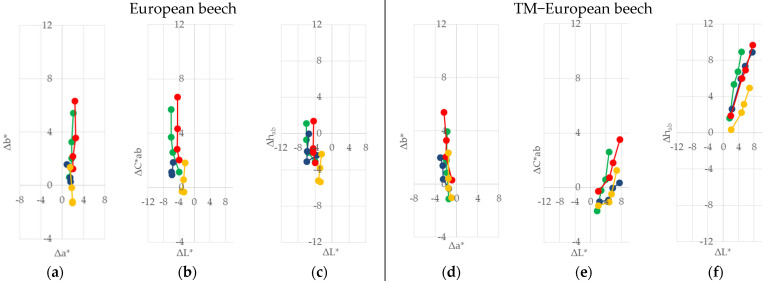
Changes in ∆*L**, ∆*a**, ∆*b**, ∆*C*_ab_*, and ∆*h_ab_* after the exposure to indoor sunlight of coated and uncoated native (**a**–**c**), and thermally treated (**d**–**f**), European beech wood for 60 days.

**Figure 4 polymers-13-03373-f004:**
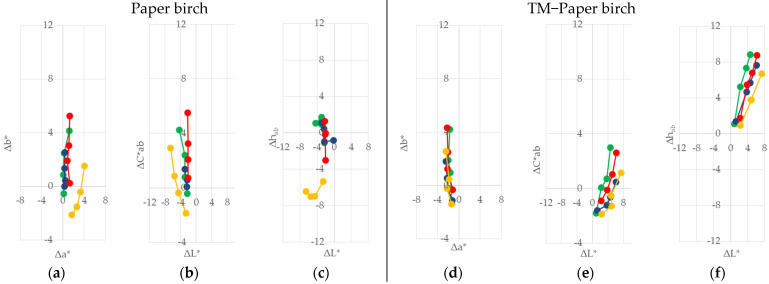
Changes in ∆*L**, ∆*a**, ∆*b**, ∆*C*_ab_*, and ∆*h_ab_* after the exposure to indoor sunlight of coated and uncoated native (**a**–**c**), and thermally treated (**d**–**f**), Paper birch wood for 60 days.

**Figure 5 polymers-13-03373-f005:**
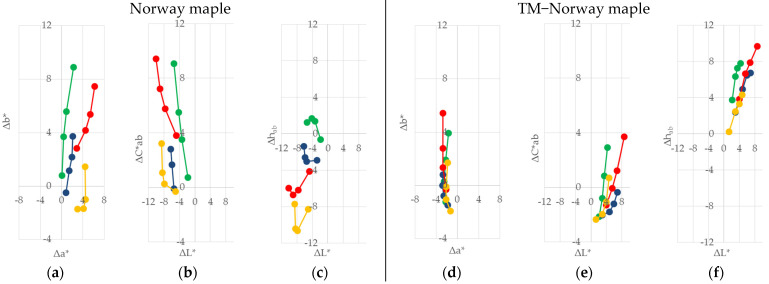
Variations in ∆*L**, ∆*a**, ∆*b**, ∆*C*_ab_*, and ∆*h_ab_* after the exposure to indoor sunlight of coated and uncoated native (**a**–**c**), and thermally treated (**d**–**f**), Norway maple wood for 60 days.

**Figure 6 polymers-13-03373-f006:**
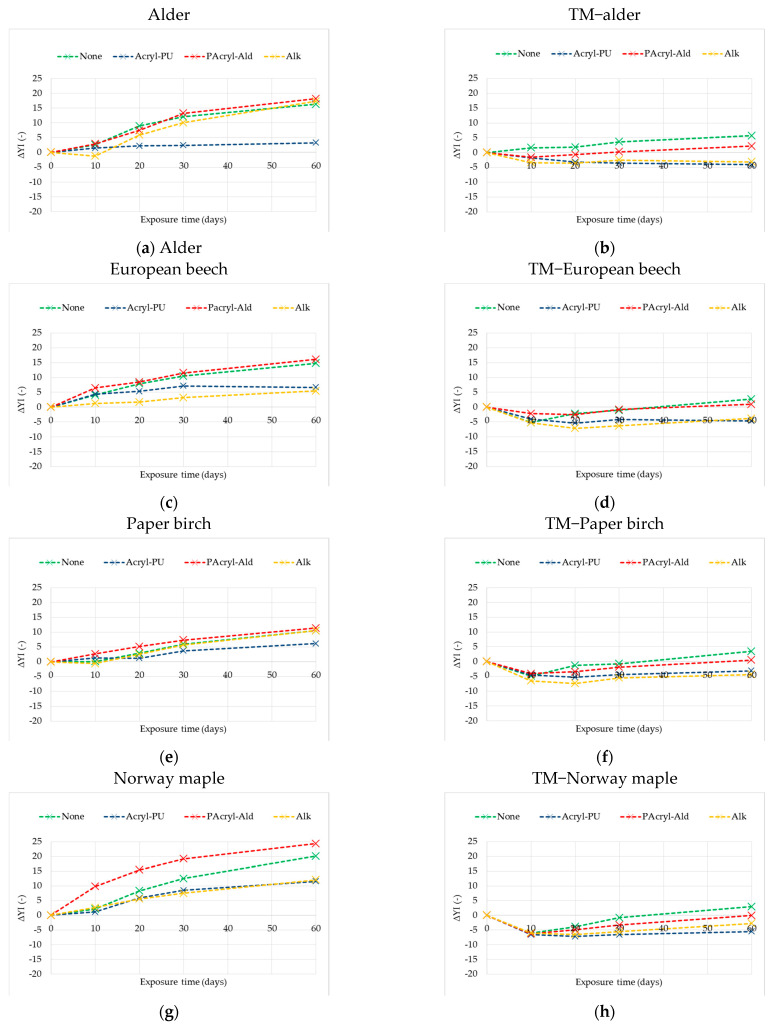
Variations in the difference in the yellowness index ∆*YI* after exposure to indoor sunlight of coated and uncoated native (**a**,**c**,**e**,**g**), and thermally treated (**b**,**d**,**f**,**h**), wood for 60 days.

**Figure 7 polymers-13-03373-f007:**
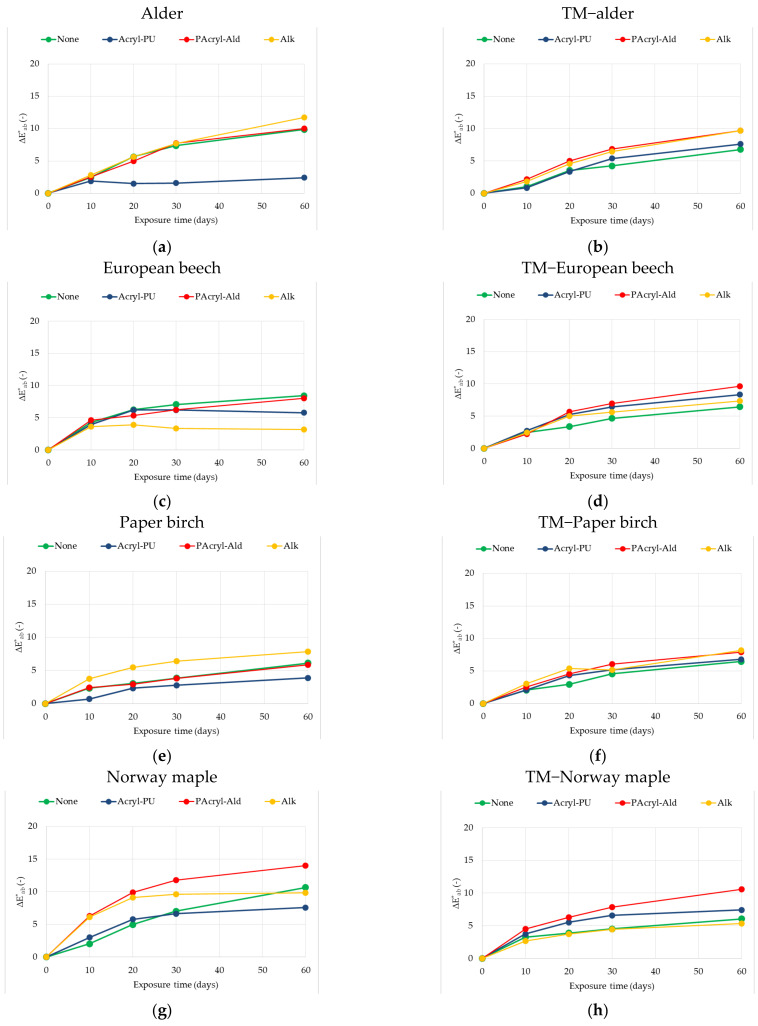
Variation in total colour difference ∆*E*_ab_* after the exposure to indoor sunlight of coated and uncoated native (**a**,**c**,**e**,**g**), and thermally treated (**b**,**d**,**f**,**h**), wood for 60 days.

**Table 1 polymers-13-03373-t001:** Characteristics of finish products and their applications.

Finish Product	Acryl-PU	PAcryl-Ald	Alk
Commercial name	Aqua TL-412 -Treppenlack/50 ^b^	PUR SL-212 -Schichtlack/30 ^b^	HWS-112 -Hartwachs-Siegel/clear ^b^
Components	One-component	Two-component	One-component
Film former	Acrylic/Polyurethane dispersion	Polyacrylic resin, Aldehyde resin	Alkyd resin
Gloss	Silk gloss	Semi-matt	Cloth matt
Density at 20 °C (g·cm^−3^)	1.03	0.94	0.88
Flow time (s)	Approx. 35 (in DIN 6 cup)	Approx. 30 (in DIN 4 cup)	Approx. 22 (in DIN 4 cup)
Spread rate (m·Lm^−2^)	100–150	80–120 per coat	1st coat: 70; 2nd coat: 60
VOC content (g·L^−1^) ^a^	<140	-	<500
Production of the mixture		Mixing with PUR H-280 100:10 by weight	
Substrate requirements:	Clear, dry, free of dust, grease and loose substances		
-Moisture content (%)	8–12		
-Sand grit	P 100–180	P 100–180	>P 180
Application/Coat number	Spray/2	Spray/2	Spray/2
Temperature of the material, air and substrate (°C)	18–25		
Intermediate sanding (Sand grit)	P 240–320		

Note: ^a^ as per the Decopaint Directive 2004/42/EG; ^b^ Finishes were supplied by the Remmers Company in Slovakia [28].

**Table 2 polymers-13-03373-t002:** Results of statistical evaluations of the colour parameters.

Factors	Lightness *L****	Redness *a****	Yellowness *b****	Chroma *C***_ab_*	Hue Angle *h_ab_*
Wood Species (WS)	0.000^a^	0.018 ^a^	0.000 ^a^	0.000 ^a^	0.000 ^a^
Wood treatments (WT)	0.000 ^a^	0.000 ^a^	0.004 ^a^	0.000 ^a^	0.000 ^a^
Finishes (F)	0.000 ^a^	0.000 ^a^	0.000 ^a^	0.000 ^a^	0.000 ^a^
Exposure Time (ET)	0.000 ^a^	0.384	0.000 ^a^	0.000 ^a^	0.000 ^a^
WS × WT	0.000 ^a^	0.358	0.741	0.546	0.000 ^a^
WS × F	0.262	0.081 ^a^	0.000 ^a^	0.000 ^a^	0.149
WS × ET	0.872	0.967	0.003 ^a^	0.131	0.001 ^a^
F × WT	0.000 ^a^	0.000 ^a^	0.004 ^a^	0.000 ^a^	0.000 ^a^
F × ET	0.577	0.520	0.000 ^a^	0.546	0.000 ^a^
WT × ET	0.008 ^a^	0.734	0.129	0.002	0.000 ^a^

Note: ^a^ statistically significant effect (*p* < 0.05).

**Table 3 polymers-13-03373-t003:** Visualization and initial colorimetric parameters of wood.

WoodSpecies	Wood Treatment	Density [4,5,6,7](kg·m^−3^)	Visual	Lightness*L****	Redness*a****	Yellowness*b****	Chroma*C***_ab_*	Hue angle*h_ab_*
Alder	Native	523(30)	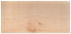	80.33(0.41)	7.36(0.31)	16.60(0.31)	18.10(0.34)	66.48(0.78)
TM	505(18)	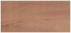	60.38(1.59)	11.36(0.50)	16.05(0.32)	19.67(0.34)	54.71(1.44)
European beech	Native	684(65)	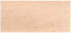	78.42(1.30)	7.33(0.52)	15.37(0.35)	17.04(0.54)	64.55(1.15)
TM	705(68)	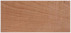	59.74(0.52)	11.82(0.22)	17.54(0.25)	21.15(0.28)	55.98(0.53)
Paper birch	Native	650(27)	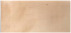	78.36(0.84)	7.14(0.29)	17.69(0.53)	19.09(0.55)	68.05(0.64)
TM	626(18)	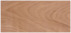	62.26(0.94)	10.95(0.34)	17.83(0.25)	20.91(0.28)	58.39(0.89)
Norway maple	Native	600(50)	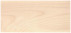	81.27(0.77)	6.00(0.33)	15.00(0.35)	16.20(0.26)	69.64(0.97)
TM	577(25)	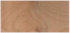	65.42(1.01)	10.49(0.35)	19.38(0.40)	22.02(0.40)	61.58(0.84)

Note: Mean values are calculated from 10 measurements. Standard deviations are in parentheses.

## Data Availability

Data sharing is not applicable to this article.

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
