# Peer review of "Colour Stabilisation of Surface of Four Thermally Modified Woods with Saturated Water Vapour by Finishes"

_polymers, 2021, doi:10.3390/polym13193373_

Round 1

Reviewer 1 Report

The manuscript concerns colour stabilisation of surface of thermally modified woods. The manuscript is interesting, good prepared. Experiments, analyzes and statistical analysis well planned and executed.

At the end of the Introduction, the Authors mention that: “…study follows the findings of the previously published studies…”. This element requires a more detailed explanation. Is the manuscript supposed to summarize/compare previous research results? What is the novelty proposed in the manuscript? What problem does this work solve?

Formula (6) concerning the calculations of the yellowness index YI require clarification of the meaning of individual components. In the remaining formulas, the meanings are described in the text or previously indicated, but this time there is no reference.

In addition, Line 123, in the above formulas the Authors use index "1" and not "2", please correct. Lines 129 and 134, wrong order of references. First appears [31], later [30], please correct.

I am not convinced of the correct choice of the Polymers to publish this manuscript. The manuscript is generally concerned with changing the properties of wood, and commercial polymer finishing materials have been used.

Author Response

Dear Reviewer,

Many thanks for the helpful suggestions for the manuscript.

The manuscript concerns colour stabilisation of surface of thermally modified woods. The manuscript is interesting, good prepared. Experiments, analyses and statistical analysis well planned and executed.

 At the end of the Introduction, the Authors mention that: “…study follows the findings of the previously published studies…”. This element requires a more detailed explanation. Is the manuscript supposed to summarize/compare previous research results? What is the novelty proposed in the manuscript? What problem does this work solve?

Yes, it was corrected. The mentioned articles are specifically devoted to the preparation of TM wood and the change of selected parameters during the treatment. In this study, we aim to stabilize the color with available commercial coatings. It is correct, as we worked with the given material to mention the work of colleagues in this form.

Formula (6) concerning the calculations of the yellowness index YI require clarification of the meaning of individual components. In the remaining formulas, the meanings are described in the text or previously indicated, but this time there is no reference.

 Yes, it was corrected.

In addition, Line 123, in the above formulas the Authors use index "1" and not "2", please correct. Lines 129 and 134, wrong order of references. First appears [31], later [30], please correct.

 Yes, it was corrected. And also in References

I am not convinced of the correct choice of the Polymers to publish this manuscript. The manuscript is generally concerned with changing the properties of wood, and commercial polymer finishing materials have been used.

This article is planned to be published in the special issue „Durability and Modification of Wood Surfaces“. The article has been notified in advance to the editorial board and it was accepted by them.

Best regards, authors.

Reviewer 2 Report

The manuscript describes the effects of four polymer coatings to the color stabilization of four varieties of untreated and thermally treated woods. It is interesting but is marginally inside the journal scope. Minor revisions are needed. I strongly suggest resubmission to the more suitable journal such as coatings.

1) Page 1 Lines 31-33 “The thermally-based modification treatment is also frequently accompanied by chemical  reactions of the cell-wall constituents cause the colour changing of wood.” Please rephrase.

2) Please give some explanations in Table 1 concerning the characteristics of finish products. For example: What is the difference between characteristic and typical odour and why this is important? Define Runout time and Application Rate.

3) I suppose that T in the last row of Table 2 must be replaced by WT.

4) The conclusions section is far too short. The authors should cite in detail the advantages and disadvantages of each polymer and their suitability/ unsuitability for each type of wood.

  Author Response

Dear Reviewer,

Many thanks for the helpful suggestions for the manuscript.

(x) Moderate English changes required

  • Yes, we tried to improve the English (yellow)

The manuscript describes the effects of four polymer coatings to the color stabilization of four varieties of untreated and thermally treated woods. It is interesting but is marginally inside the journal scope. Minor revisions are needed. I strongly suggest resubmission to the more suitable journal such as coatings.

1) Page 1 Lines 31-33 “The thermally-based modification treatment is also frequently accompanied by chemical  reactions of the cell-wall constituents cause the colour changing of wood.” Please rephrase.

Yes, it was corrected. The new sentence is „The thermally-based modification treatment is also accompanied by chemical reactions of the cell-wall components (polysaccharides, lignin and extractives) that cause the colour of the wood to change.

2) Please give some explanations in Table 1 concerning the characteristics of finish products. For example: What is the difference between characteristic and typical odour and why this is important?

The odour characteristics of finish products are noted in their Technical Data Sheet. All documents are available online: https://en.remmers.com.  Yes, we accept this comment. We evaluated cured paints. From this point of view, the odour of the paint substance in the liquid state is negligible in the characteristics of the overall surface treatment.

Define Runout time and Application Rate.

Yes, it was corrected with: „Flow time“and „Spread rate“

3) I suppose that T in the last row of Table 2 must be replaced by WT.

Yes, it was corrected with: WT = Wood treatments.

4) The conclusions section is far too short. The authors should cite in detail the advantages and disadvantages of each polymer and their suitability/ unsuitability for each type of wood.

Yes, a conclusion has been added to this proposed aspect.

Best regards, authors.